# From iron curtain to green belt: Shift from heterotrophic to autotrophic nitrogen retention in the Elbe River over 35 years of passive restoration

Alexander Wachholz[1], James W Jawitz[2], Dietrich Borchardt[1]

[1]Department of Aquatic Ecosystem Analysis and Management, Helmholtz-Centre for Environmental Research (UFZ), Magdeburg, Germany

[2] Soil and Water Sciences Department, University of Florida, Gainesville, FL, USA

*Correspondence to*: Alexander Wachholz (alexander.wachholz@uba.de)

**Abstract.** We investigate changes in in-stream nitrogen retention and metabolic processes in the River Elbe between 1978 and 2020. We analyzed multi-decadal time series data and developed a metabolic nitrogen demand model to explain trends in dissolved inorganic nitrogen (DIN) retention, gross primary production (GPP), and ecosystem respiration (ER) during a

period of highly dynamic pollution pressures in the Elbe River (Central Europe). Our findings reveal a marked increase in summer DIN retention and a decrease in winter DIN retention, establishing a distinct seasonal pattern. We identified three periods in the Elbe's DIN retention dynamics: dominantly heterotrophic under high organic and inorganic pollution pressure (1980-1990), transition (1990-2003), and dominantly autotrophic with lower pollution (2003-2017). We link these changes to reduced industrial pollution, improved wastewater treatment, and a shift in the in-stream balance between heterotrophic

and autotrophic processes. During the first period, high ER  caused  elevated metabolic nitrogen demands, primarily driven by heterotrophic processes. As pollution from industrial and wastewater emissions decreased, GPP rates increased, and ER gradually declined, prompting a shift towards an autotrophic-dominated nitrogen retention regime. Our study indicates a tight coupling of nutrient reduction from external sources and dominant processes of natural attenuation in large rivers, which needs to be considered for projections of recovery trajectories toward sustainable water quality.

## 1 Introduction

Large river systems have been substantially impacted by anthropogenic pressures associated with economic development, including observed long-term trends of ecosystem degradation throughout much of the 20th century (Vörösmarty et al., 2015; Meybeck et al., 2018). However, ecosystem protection regulations promulgated in recent decades have supported the recovery of many river ecosystems (Minaudo et al., 2015; Westphal et al., 2019), including improvements in river metabolic

regimes (Diamond et al., 2022; Jarvie et al., 2022) and reduction of dissolved inorganic nitrogen loads (Wachholz et al., 2022). Much work has examined the terrestrial drivers of these multi-decadal trajectories of river ecosystems (Ehrhardt et al., 2019; Dupas et al., 2018; van Meter et al., 2017). However, little is known about the impact of those long-term changes on the in-stream processes. An important in-stream process that is susceptible to external pressures is in-stream retention of dissolved inorganic nitrogen (DIN), which plays a crucial role in watershed nitrogen (N) budgets. On the global scale, inland

waters retain 60% of the annual terrestrial nitrogen input (Schlesinger and Bernhardt, 2013). This important ecosystem function also helps protect downstream ecosystems from eutrophication (Bianchi et al., 2010) induced by highly reactive

forms of nitrogen, such as nitrate and ammonium (Seitzinger et al., 2002). In-stream DIN retention is closely linked to other ecosystem functions, especially stream metabolism (Hall and Tank, 2003; Heffernan and Cohen, 2010). Long-term changes of metabolism (Arroita et al., 2019), nitrogen loading (Ballard et al., 2019), and nitrogen composition (Wachholz et al., 2022) have been observed in rivers. Here, we are interested in associated long-term patterns of in-stream DIN retention.

Here we use in-stream DIN retention as an overarching term for all processes that, temporarily or permanently, remove DIN from the water. These processes include assimilation by hetero- and autotrophic organisms, and denitrification. In-stream DIN retention is performed by bacteria, algae and macrophytes in the water column and sediments (Deutsch et al., 2009; Middelburg and Nieuwenhuize, 2000), which either assimilate DIN into their biomass or use it for metabolic processes. The activity of these organisms is influenced by various environmental factors such as water temperature, residence time, and nutrient concentrations (Collos and Harrison, 2014; Rasmussen et al., 2011; Snell et al., 2019). While travel time (Bertuzzo et al., 2017) and water temperature (Sherman et al., 2016) are often assumed to be the primary controls of in-stream DIN retention, the composition of DIN can also play a significant role. For instance, $NH_4$-N is favored over $NO_3$-N by many algae and bacteria (Cejudo et al., 2020). If sufficient $NH_4$-N is available, the DIN uptake by unicellular algae can increase by a factor of 2-16 (Collos and Harrison, 2014). Understanding of DIN retention should also consider the relative contributions of algae, bacteria, and macrophytes, which each have preferences for different DIN species (Bergbusch et al., 2021; Collos and Harrison, 2014), incorporate N at different stoichiometric ratios into their biomass (Diamond et al., 2022; Godwin and Cotner, 2018), and have different growth efficiencies (the ratio of consumed resources that are assimilated into biomass, e.g. del Giorgio, 1997).

While long-term trends in the drivers and correlates with in-stream DIN retention are relatively well known (e.g, Ballard et al., 2019; Wachholz et al., 2022, Diamond et al., 2022), their consequences on in-stream DIN retention itself are understudied. This leaves considerable uncertainty in long-term watershed N budgets, which are already uncertain due to hard-to-quantify phenomena, such as time lags (Lutz et al., 2022). Therefore, we propose the following research question: How do the magnitude and dominant processes of in-stream DIN retention change in response to long-term changes in DIN composition and river trophic regime? To answer this question, we studied the Elbe River from 1978 to 2017. During this period, the Elbe river underwent a significant transition: before 1990, most of its catchment lay beyond the iron curtain and experienced significant chemical pollution, especially from heavy fertilizer use due to the agro-industrial revolution in the German Democratic Republic (GDR) after the 1960s (Bauerkaemper, 2004). Furthermore, large amounts of untreated wastewater from urban and industrial areas further polluted the stream (Netzband et al., 2002). However, after the GDR's collapse in 1989, industrial facilities closed, and WWTPs were rapidly constructed in the 1990s following the German reunification in 1991, resulting in decreased emissions from these sources and improved water quality in the Elbe (Adams et al., 2001). Parts of the Elbes remaining floodplain are now located within the European Green Belt with the aim to preserve its valuable functions for flood retention and biodiversity (Serra-Llobet et al., 2022).

Previous work suggests a shift from a heterotrophic (primarily production / respiration < 1) to an autotrophic system (primarily production / respiration > 1; Doretto et al., 2020) following the reduction of riverine biological oxygen demand in response to wastewater treatment improvements following the GDR collapse (Lehmann and Rode, 2001). However, the concomitant changes in DIN retention during this period have not been examined.

Quantitative links between in-stream metabolism and nutrient retention have been described by many authors (Hall and Tank, 2003; Heffernan and Cohen, 2010; Kamjunke et al., 2021; Rode et al., 2016; Zhang et al., 2023). The N demand of gross primary production (GPP) and ecosystem respiration (ER) in an ecosystem can be estimated, subject to assumptions about growth efficiencies (the share of GPP/ER that leads to biomass growth), biomass C:N ratios, and photosynthetic/respiratory quotients ($O_2$/C ratio during photosynthesis/respiration) (Hall and Tank, 2003). As both auto- and heterotrophic microorganisms use DIN as their preferred N source (Rier and Stevenson, 2002) those assumptions allow us to link the dissolved oxygen (DO) and the DIN balance of a river segment. Other processes, however, disturb this link by influencing either DO or DIN but not both. Possible examples are other biologic processes such as nitrification (retains DO but does not affect DIN budget directly) and denitrification (removes $NO_3$-N but does not consume DO). However, physicochemical effects such as ad- or desorption of $NH_4$-N also influence the DIN budget of a river segment without affecting the DO budget (Triska et al., 1994).

GPP in the Elbe is mostly caused by phytoplankton (Hardenbicker et al., 2014), and phytoplankton activity is closely linked to the in-stream N processes in the Elbe in recent years (Ritz and Fischer, 2019; Kamjunke et al., 2021). In-stream denitrification is assumed to be of lesser importance in the Elbe, at least after the reunification (Ritz et al., 2017; Schulz et al., 2023). The strong decrease in $NH_4$-N concentrations after 1990 (Adams et al., 2001) suggests that nitrification and sorption processes play a minor role in the Elbe's DIN retention, as less $NH_4$-N is available. Before 1990, in-stream oxygen concentrations were low (Lehmann and Rode, 2001), and $NH_4$-N concentrations were high (Adams et al., 2001). We, therefore, expect that nitrification and denitrification occurred at relatively higher rates with weaker coupling between DIN retention and metabolic processes.

To summarize, we hypothesize a strong coupling between metabolic processes and in-stream DIN retention (Heffernan and Cohen, 2010). We predict this coupling to be weakened during periods of high pollution (such as during the late phases of the GDR before 1990) when processes such as denitrification, nitrification, and sorption influence the DO and N cycles independently.

To test our hypothesis, we quantified DIN retention using a two-station mass balance approach along an 110 km, 8th-order segment of Elbe with no noteworthy tributaries. Furthermore, we quantified changes in the trophic regime in the Elbe by estimating gross primary production (GPP) and ecosystem respiration (ER) using the single-station hourly oxygen mass-balance approach (Odum, 1956). We linked in-stream DIN retention to metabolic rates using stoichiometric constraints and assessed the relative importance of autotrophic and heterotrophic processes.

## 2 Data and methods

### 2.1 Overview

We used DIN and DO mass balances to assess the magnitude and responsible processes of in-stream DIN retention over 42 years from 1978 to 2020. We inversely modeled DO concentrations using a maximum likelihood estimation method to quantify in-stream metabolic processes (GPP, ER) over time. We linked metabolism and in-stream DIN retention using a simple model based on stoichiometric constraints and growth efficiencies.

### 2.2 Study site

We studied the last 111 km of a 1094 km 8th-order river (Elbe, Fig.1a, b) between Schnakenburg (km 474) and Geesthacht (km 585) which was a part of the Iron Curtain before 1990. The German Elbe is free flowing, meaning that at least 474 km upstream of the studied segment, no dams are present. The studied segment has no noteworthy tributaries and is located 30 km downstream of the last larger tributary (Havel), contributing between 10 and 20% of the Elbe discharge (Fig. S1). The annual mean discharge at the downstream station Geesthacht (km 585) is 716 $m^3s^{-1}$ (IKSE, 2005). Lateral groundwater discharge was estimated to be <0.05 $m^3$ $s^{-1}$ $km^{-1}$ for the Elbe at stream km 450 (Zill et al., 2023), which corresponds to 5.5 $m^3$ $s^{-1}$ for the entire segment, less than 1% of the annual mean discharge, and was therefore not considered. We estimate the depth of the segment to be between 2.8 and 4.5 m and the width between 270 and 450 m during flow conditions between the 5th and 95th percentile (see Section 2.5 for details). PO4-P concentrations oscillated between 0.2 and 0.3 mg $l^{-1}$ before 1990 and between 0 and 0.1 mg $l^{-1}$ afterward (Wachholz et al., 2024). The aquatic productive season occurs between April and October, as indicated by increased Chlorophyll-a concentrations, and without macrophytes, primary production in the Elbe is assumed to be caused by phytoplankton (Hardenbicker et al., 2018).

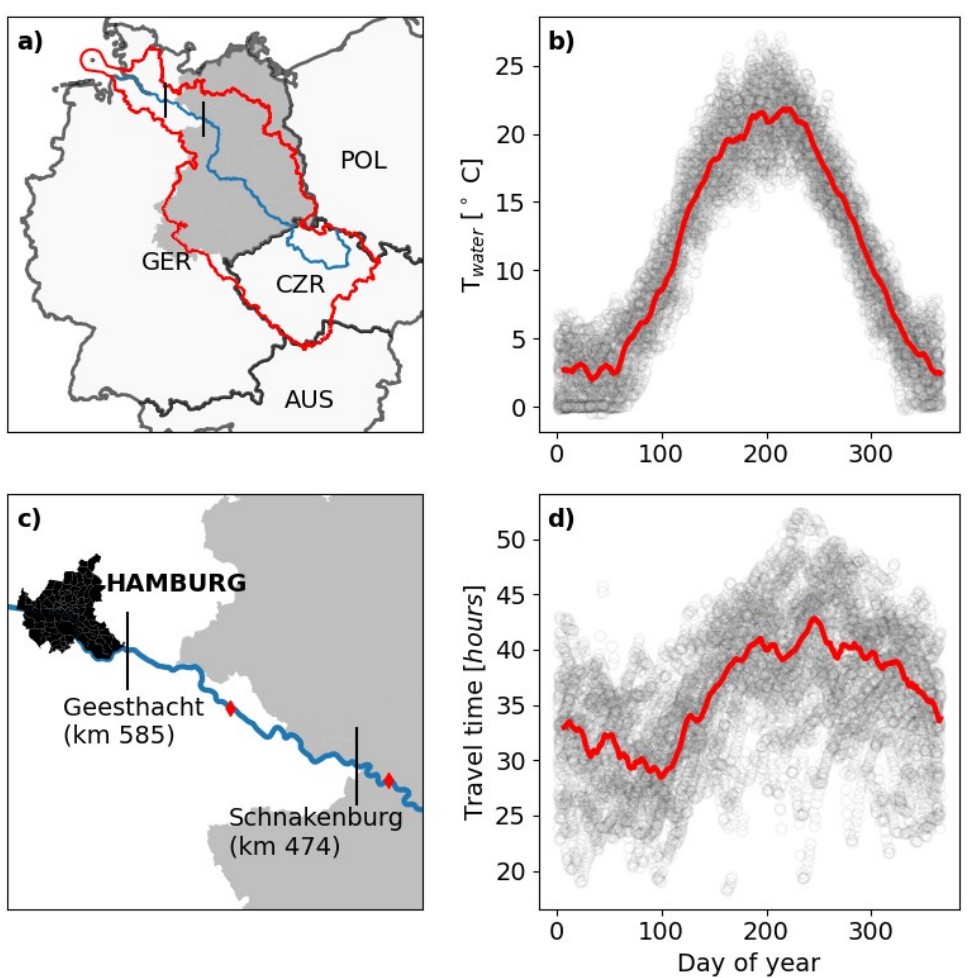


**Figure 1: Location of the investigated river segment (a, c) in Europe. The blue line marks the Elbe main stream and the red outline represents the catchment. Black vertical lines mark the beginning and the end of the segment (sampling locations). Red diamonds in panel c indicate discharge gages. The former area of the German Democratic Republic (GDR) is represented by the grey shade. Intra-annual patterns of water temperature (b) and travel time (d).**
**Circles show raw data, and red lines multi-year 7 day running means for each day of the year.**

**2.3 Two-station mass-balance**

To quantify in-stream DIN retention we use a two-station mass balance approach (e.g., Ritz and Fischer, 2019). All samples were collected and analysed by the federal state authority of Lower Saxony. Nitrate samples were analysed with liquid ion-

chromatography (DIN EN ISO 10304-1) and ammonium with flow analysis and spectrometric detection (DIN EN ISO 11732). It is not documented if and how the detection method changed over the monitored period. However, the time series

exhibited no obvious break points (Fig. S2). The upstream site (Schnackenburg) is located at stream km 474 and has weekly (1978-1991) and bi-weekly (1991-2021) water quality ($NO_3$-N and $NH_4$-N) time series available. Nitrite ($NO_2$-N) was not considered as, even in high nitrogen pollution periods, it made up between 1 and 3% of the DIN for both input and output
(Fig. S3). The upstream discharge gage (Wittenberge) is located 21 km upstream of the sampling site (Fig. 1c). The downstream site (Geesthacht) at stream km 585 has weekly $NO_3$-N and $NH_4$-N from 1980-1988, bi-weekly until 2006, and monthly afterward. A data gap from 1988-1993 for the downstream site was filled with data from another sampling site located at stream km 598 (Zollenspieker), as both sites exhibit very similar nitrogen concentrations (Wachholz et al. 2022, their Figure S2). The downstream discharge gage (Neu Darchau) is located 50 km upstream of the sampling site used to
estimate DIN load (station Geesthacht, Fig. 1c). However, the difference in catchment area between the sampling site and the gage is less than 3% (Wachholz et al., 2022). We consider this by assuming a 10 % error in discharge measurements in our uncertainty propagation, while a previous mass balance study (Ritz and Fischer, 2019) reported errors $\leq$ 5% in the Elbe, according to personal communication with the state authorities responsible for the measurements. The same study also reported an error of 10 % for the laboratory quantification of nitrogen compounds according to the German standard
procedures, which were also used in this study.

### 2.3.1 WRTDS model

To reconcile differences in sampling dates between the upstream and downstream stations, daily loads of $NO_3$-N and $NH_4$-N were estimated using the weighted regression on time, discharge, and season (WRTDS) function of the R package EGRET (Hirsch et al., 2010). The WRTDS function uses a weighted regression approach to estimate daily loads and concentrations,
accounting for non-linearity and non-stationarity in the relationships between the time, discharge, season, and concentrations over time. Measured and simulated daily concentrations showed very good agreement ($R^2 > 0.7$ and percent bias < 3, see Fig. S4).

### 2.3.2 Retention metrics

Using the daily loads provided by the WRTDS function, we calculated in-stream DIN retention $R_{obs}$ as

$$R_{obs} = L_{in} - L_{out} \qquad (I)$$

where L is the DIN load [kg day$^{-1}$]. Uncertainty in $R_{obs}$, was computed based on Gaussian error propagation (Section S2). We then calculate the relative retention $RR$

$$RR_{obs} = \frac{R_{obs}}{L_{in}} \qquad (II)$$

and the area weighted retention $U$ [kg d$^{-1}$m$^{-2}$]

$$U_{obs} = \frac{R_{obs}}{A} \qquad \qquad \text{(III)}$$

where A [$m^2$] is the bottom area of the river segment (Stream Solute Workshop, 1990). We calculate Robs, RR$_{obs}$, and U$_{obs}$ for DIN. We corrected the L$_{out}$ time series for travel time to align the inflow and outflow time series. The estimated travel times for the segment ranged from 19 and 52 hours (Fig. 1d), but since loads were only available as daily means, we evaluated corrections in increments of single days (Table S1), and found that shifting L$_{out}$ one day ahead of the L$_{in}$ series led to the best fit between the inflow and outflow time series.

## 2.4 Channel geometry estimations

Calculating area-weighted retention rates and inverse modeling of in-stream metabolism for a river segment requires knowledge of the surface area, water residence times, and channel depth. The methods to obtain these estimates are summarized below but are described in detail and validated in Section S1 and Fig. S5. We used discharge-based transfer functions to estimate the geometrical parameters at different water levels (e.g., Booker and Dunbar, 2008). We estimated travel time τ using a transfer function proposed by (Scharfe et al., 2009) for the German Elbe. For the channel area of the investigated segment, we established a transfer function based on discharge, which we parametrized with channel areas derived at different discharge conditions from Sentinel 2 images with a surface water detection algorithm (Normalized Difference Water Index). We employed a power law model for the channel depth based on data from Aberle et al. (2010).

## 2.5 Metabolism model

The single station method for metabolism estimation in rivers uses DO data from a single monitoring station to estimate the rates of gross primary production (GPP) and ecosystem respiration (ER) in the river (e.g., Hall et al., 2016):

$$DO_{t+\Delta t} = DO_t + \frac{GPP}{PPFD_{24}} PPFD_t - ER \, \Delta t + \frac{k \, 600 \left(\frac{Sc}{600}\right)^{\left(\frac{-1}{2}\right)} \left(DO_{max} - DO_t\right)}{z} \Delta t \qquad \text{(IV)}$$

where *GPP* and *ER* are the daily rates [mmol $m^{-3}$ $d^{-1}$] of the respective parameters, *PPFD$_{24}$* is the photosynthetic photon flux density for the day and *PPFD$_t$* per hour[µmol $m^{-3}$ $d^{-1}$], *k600* is the gas exchange rate [m $d^{-1}$], *Sc* is the dimensionless Schmidt number for oxygen (Wanninkhof, 1992) which is calculated based on water temperature, *DO$_t$* and *DO$_{max}$* are the actual and maximum (at 100% saturation) *DO* concentrations [µmol $l^{-1}$], and *z* is the channel depth [m].

### 2.6 Data preparation and metabolism estimation

Implementation of Eq. IV requires hourly estimates of *PPFD*, *DO*, *DO*$_{max}$, *z*, and *Sc*. We interpolated diurnal DO concentrations by fitting sine functions to a time series of daily mean, minimum, and maximum values from 1978 to 2017, as described in Section S3. Simulated DO concentrations were validated with two years of hourly measured values showing characteristics of a good fit ($R^2$=0.96, RMSE=0.42 mg l$^{-1}$). We estimated hourly solar radiation (as *PPFD*) based on the method proposed by Duffie and Beckman (2013), which is implemented in the Python package *solarPy*. We calculated DO saturation based on the method of Weiss (1970) using hourly air pressure data from the German weather service (DWD) station Seehausen (ID 4642) and daily mean water temperature measured together with the DO data. We estimated k600 for the segment with a hydraulic equation from Raymond et al. (2012, their Equation 7 of Table 2; see Section S4 for details) and we used maximum likelihood estimation (e.g., van de Bogert et al., 2007) to fit parameter distributions for *GPP* and *ER* for each day of the time series. We used a limited memory Broyden–Fletcher–Goldfarb–Shanno algorithm, implemented in the Python package *Scipy* (Virtanen et al., 2020; function minimize(method='L-BFGS-B')) to minimize the negative log-likelihood between the modeled DO concentrations and the observed DO data (e.g., van de Bogert et al., 2007). We constrained daily *GPP* and *ER* estimates to be between 0 and (-) 50 [g O$_2$ m$^{-2}$ d$^{-1}$] and use (-)10 [g O$_2$ m$^{-2}$ d$^{-1}$] as an initial estimate.

Especially the use of estimated *k600* and the daily mean water temperature might introduce large uncertainties to our *GPP* and *ER* estimations. To quantify these uncertainties, we estimated the daily standard deviations σ of *k600* (based on the parameter variability described in Raymond et al., 2012), the daily water temperature (based on hourly data which are available some years), and *DO*$_{max}$ (based on the propagation of the standard deviations from water temperature) for each day (see Section S4). We performed 100 bootstrap iterations for each day, drawing errors from normal distributions N~(0, σ) for the parameters *k600, Do*$_{max}$, and daily water temperature. We reported the mean and standard deviations of the resulting *GPP* and *ER* estimations. We assessed the goodness of fit for the *GPP* and *ER* estimations by simulating DO concentrations using the mean daily *GPP* and *ER* rates and comparing those to observed DO concentrations, reporting daily $R^2$ and root mean square error values (Fig. S10). We further assessed the effects of the uncertainty of *k600, Do*$_{max}$ and daily water temperature on *GPP* and *ER* estimated by performing a simplified one-factor-at-a-time sensitivity analysis for two exemplary years (see Fig. S11). The model implementation in Python can be found at (github.com/alexiwach/MetabolismModelElbe). We also assessed the areal extent of the metabolic signal and found it in line with the mass balance analysis (see Section S4).

### 2.7 Estimating the N demand of metabolic processes

The demand of N caused by GPP and ER can be estimated based on the respective organisms' growth efficiency (*GE*), the photosynthetic- and respiratory quotient (*PQ*, *RQ*), and the C:N ratio of their biomass (e.g., Hall and Tank, 2003). *GE* [-]

describes the proportion of resources and energy that is captured by photosynthesis ($GE_{AUTO}$) or respiration ($GE_{HET}$) that is incorporated into new biomass (del Giorgio et al., 1997; Hall and Tank, 2003). *PQ* and *RQ* describe the ratio of $O_2$ produced/

consumed per $CO_2$ consumed/ produced (Berggren et al., 2012; Hall and Tank, 2003). Those two concepts can be used to assess how much C is incorporated into biomass for any given GPP and ER rate. Via the C:N ratio of the biomass (C:$N_{HET}$, C:$N_{AUT}$), the N demand can then be estimated. Since autotrophic and heterotrophic microbes prefer DIN to other forms of nitrogen (Rier and Stevenson, 2002) and DIN is always available at concentrations > 1 mg l$^{-1}$, we interpret the N demand as DIN demand.

For autotrophic processes, we can formulate the following

$$U_{AUT}(t) = GPP(t) \frac{1}{PQ} \frac{GE_{AUT}}{C:N_{AUT}}$$
(V)

where $U_{AUT}$ [mol m$^{-2}$ day$^{-1}$] is the DIN demand of the GPP rate [mol m$^{-3}$ day$^{-1}$], and *PQ* [mol C mol$^{-1}$ $O_2$], $GE_{AUT}$ is the

autotrophic growth efficiency [-], $C:N_{AUT}$ is the carbon to nitrogen ratio in the autotrophic biomass and *z* is the channel depth [m] used to convert volumetric GPP to areal retention rates. Similarly, we can formulate for heterotrophic processes

$$U_{HET}(t) = R_{het}(t) RQ \frac{GE_{HET}}{C:N_{HET}}$$
(VI)

however, it is well known that the ER is not only caused by heterotrophic bacteria, but also by autotrophs (e.g. Hall and Tank, 2003). A way to correct autotrophic respiration is to subtract the non-biomass-producing fraction of GPP from ER (Hall and Tank, 2003).

$$R_{het}(t) = ER(t) - GE_{AUT} GPP(t)$$
(VII)


combining Eq. V, VI, and VII allows us to estimate the complete metabolic N demand $U_{met}$ as follows

$$U_{met}(t) = \left( GPP(t) \frac{1}{PQ} \frac{GE_{AUTO}}{C:N_{AUTO}} + \left( ER(t) - GE_{AUTO} GPP(t) \right) RQ \frac{GE_{HET}}{C:N_{HET}} \right) z(t)$$
(VIII)

Calculating $U_{met}$ based on GPP and ER rates, therefore, requires estimations of six parameters. For C:$N_{AUT}$, we used 7.3, which was reported for phytoplankton biomass in the Elbe and we used *PQ* = 1 [mol C mol$^{-1}$ $O_2$] and $GE_{AUTO}$ = 0.5, which have been successfully used to explain assimilatory DIN uptake in the Elbe (Kamjunke et al., 2021). It is well known that

the parameters $C:N_{HET}$, $GE_{HET}$, and $RQ$ show strong variability across ecosystems (Godwin and Cotner, 2018; del Giorgio and Cole, 1998; Berggren et al., 2012). Since they all affect $U_{met}$, we evaluated three parameter combinations that would lead to

low, intermediate, and high $U_{met}$ values (Table 1).

**Table 1: Parameter combinations for the estimation of the metabolic nitrogen demand. $C:N_{HET}$ is the carbon to nitrogen ratio of heterotrophic organisms, $GE_{HET}$ is the heterotrophic growth efficiency [-] and $RQ$ is the respiratory quotient [mol $O_2$ mol$^{-1}$ C].**

| Parameter combination | $C:N_{HET}$ | $GE_{HET}$ | $RQ$ |
|---|---|---|---|
| Source | Godwin and Cotner (2018) | del Giorgio and Cole (1998) | Berggren et al (2012) |
| $U_{METlow}$ | 8 | 0.04 | 0.8 |
| $U_{METmean}$ | 5 | 0.25 | 1.2 |
| $U_{METhigh}$ | 4 | 0.6 | 1.6 |

We compared $U_{met}$ estimated from each of these parameter combinations to $U_{obs}$ based on the mass balance approach. $U_{met}$, as introduced here, only considers the nitrogen demand directly associated with assimilation related to primary production or respiration. Other processes, as e.g. denitrification, are not considered but discussed later.

## 3. Results and Discussion

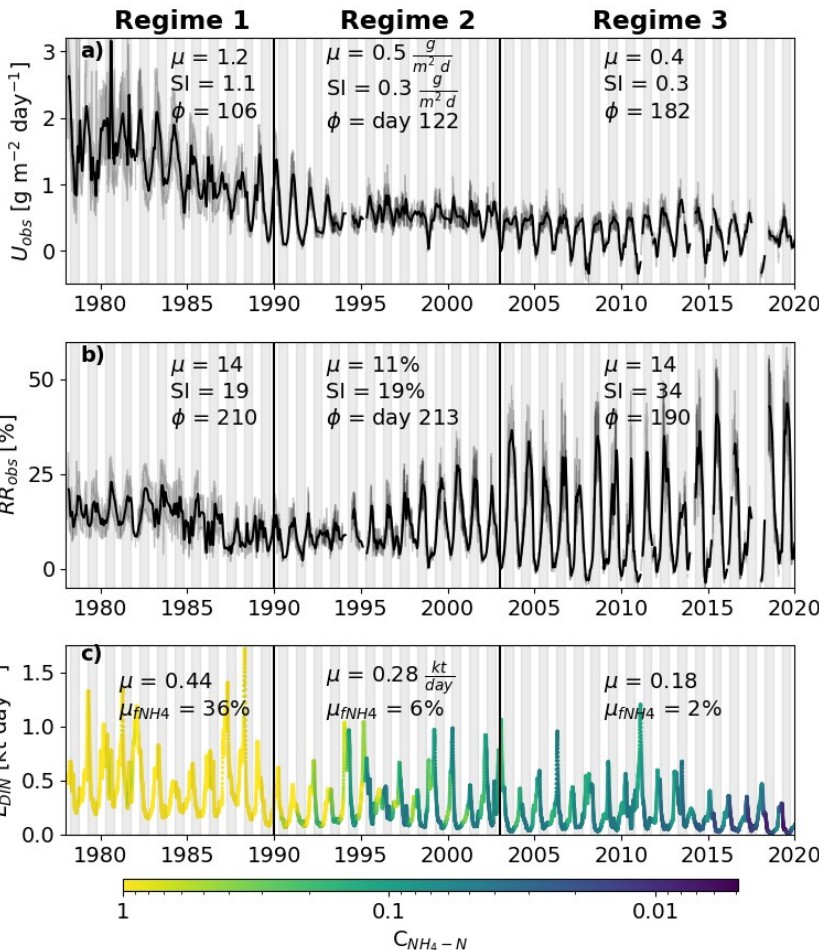

**Figure 2: Area-weighted ($U_{obs}$, panel a) and relative dissolved inorganic nitrogen (DIN) retention rates ($RR_{obs}$, panel b) estimated using a two-station mass balance approach. The three black vertical lines correspond to the major changes in U: Regime 1 shows high mean (μ), strong seasonal amplitude (SI) and peaks during spring (Φ = day 106). Regime 3 has a much lower μ and SI, while the day of the peak occurs during summer (Φ =182). Regime 2 represents the transition between both. The black lines represent a 30-day moving average value, the shaded area around the black line shows raw values with a 90 % confidence interval. Panel c) shows the DIN load (L) received by the investigated segment of the Elbe. The color represents the corresponding ammonium concentration. μ$_{fNH4}$ is the mean fraction of DIN that consists of ammonium. The white background represents the colder six month of the year (October-April) and dark background the warmer six.**

**3.1 DIN retention**

The highest $U_{obs}$ values during the entire time series were observed during regime 1 (1978-1990), oscillating between 1 and 2

g m$^{-2}$ d$^{-1}$ (Fig. 2a). Starting in mid of regime 1, $U_{obs}$ decreased and oscillated mostly between 0 and 1 for the rest of the time
series, with some negative values occurring during the winter in three years (2008, 2011, and 2018). During regime 1, $U_{obs}$
peaked shortly before the aquatic growing season (days 106-122, April-May) and showed clear summer peaks (day 182,
July) afterward. The relative retention, however, showed a consistent summer peak during the entire time series (days 190-
220, July-August) while the amplitude increased in regimes 2 and 3 (19% to 34%) (Figure 2b). Annual mean loads decreased

throughout all three regimes from 0.44 to 0.18 kt day$^{-1}$ with a substantial decrease in annual minima around 1989 (Figure 2c).
Likewise, the share of NH$_4$-N from the DIN load declined from 36 to 6%. Concentrations of NH$_4$-N did not reach values > 1
mg l$^{-1}$ later in regime 2.

**3.2 Metabolism**

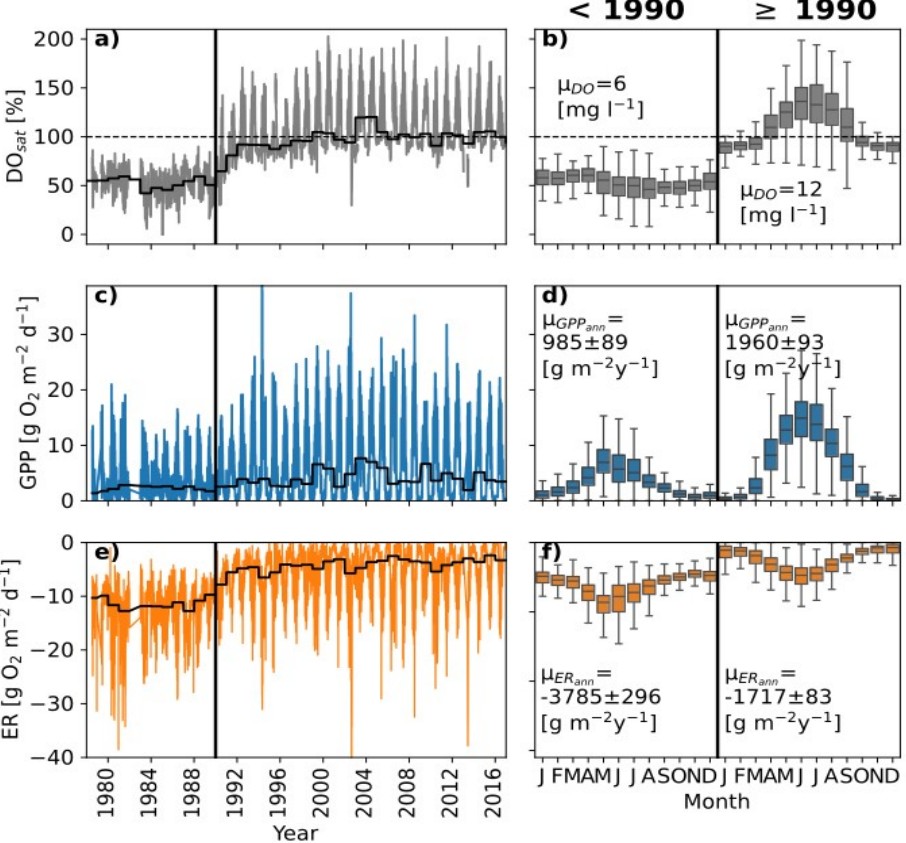

**11Figure 3. Time series of daily dissolved oxygen saturation (*DO$_{sat}$* a), daily gross primary production estimates (*GPP*, c), and ecosystem respiration estimates (*ER*, e). The black lines show the annual median *DO, GPP* and *ER* values. Panels b, d, and f show the mean monthly patterns for DO$_{sat}$, GPP, and ER, each before and after 1990. μ$_{DO}$ represents the mean DO concentrations, *μ$_{GPPann}$* and *μ$_{GPPann}$* the mean annual GPP and ER rates.**

### 3.2.1 Oxygen saturation

The multi-decadal pattern of DO in the Elbe showed distinct behavior before and after 1990, coinciding with the German reunification and the collapse of the iron curtain (Figure 3a). Oxygen saturation before 1990 oscillated between 20 and 70%, but increased rapidly after 1990, reaching super-saturation for the first time in 1991. Before 1990 there was no clear intra-annual pattern (Figure 2b), but for the rest of the time series, *DO$_{sat}$* oscillated seasonally between ~80% and ~ 180%, peaking around day 180, coinciding with the annual peaks of residence time, water temperature, and area-weighted DIN retention (Fig. 1b, c; Fig. 2a).

### 3.2.2 Gross primary production and ecosystem respiration

Similar to $DO_{sat}$, GPP showed a clear change around 1990 with low annual peaks (10-15 g $O_2$ m$^{-2}$ day$^{-1}$) before and higher (~15-25g $O_2$ m$^{-2}$ day$^{-1}$) in the following years (Fig 3c). Compared to $DO_{sat}$, the timing of the GPP peak stayed similar throughout the time series (July), Fig 3d.

While the seasonal pattern of ER stayed similar (peaks in May, June, July) throughout the time series, annual ER rates were reduced to less than half after 1990 (Fig 3e, f). Most apparent are higher ER rates during the winter months before 1990. In contrast to GPP, ER rates started to change as early as 1987.

Considering annual mean net ecosystem productivity ($NEP = GPP-ER$), the investigated segment of the Elbe is a net-heterotrophic system before 1990 which means more $O_2$ is consumed than produced. High ER rates before 1990, led to a DO deficit of ~ 2800 g $O_2$ m$^{-2}$ year$^{-1}$. After 1990 reduced ER and increased GPP rates turn the NEP of the segment and led to an oxygen surplus of ~ 240 g $O_2$ m$^{-2}$ year$^{-1}$.

310

### 3.3 Linking Metabolism and DIN retention

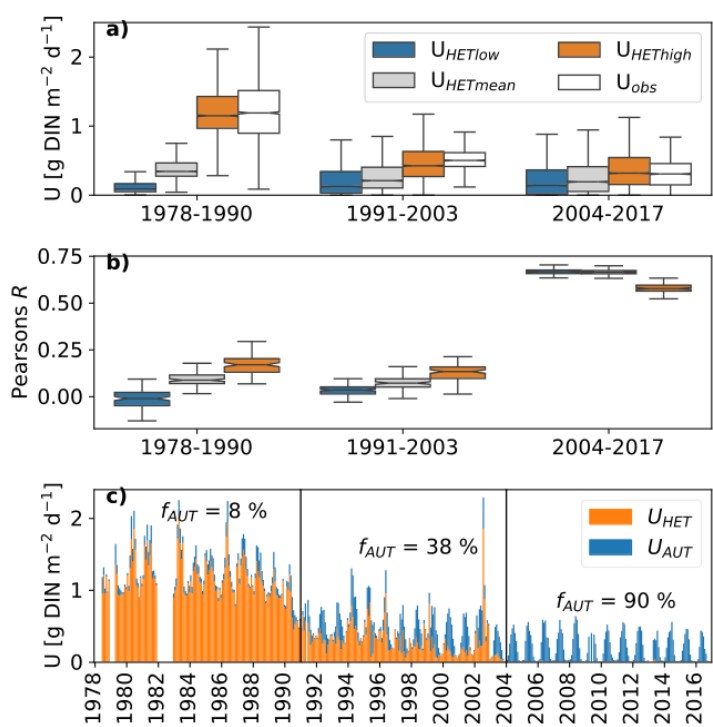

**Figure 4. (a) Simulated and observed in-stream nitrogen retentions for three periods. The different colored boxplots represent the three parametrizations of heterotrophic nitrogen demands described in Table 1.** $U_{obs}$ **represents retention rates based on the two-station mass balance described in Section 2.3. (b) Pearson correlation coefficients for the three parametrizations of heterotrophic nitrogen demands with** $U_{obs}$**. (c) Monthly mean timeseries of U, based on the best fit parameter combinations for each period:** $U_{HEThigh}$ **for period 1 & 2 and** $U_{HETlow}$ **for period 3. The colors indicate the share of the hetero- and autotrophic nitrogen demands.** $f_{AUT}$ **is the fraction of U caused by autotrophic nitrogen demand [%].**

The different parametrizations of heterotrophic nitrogen demand show diverging performances over the three investigated periods which correspond to the three regimes of $U_{obs}$ distinguished in Figure 2. Between 1978 and 2003, only the parameterization $U_{METhigh}$ led to a similar distribution of U than observed (Figure 4a). During the last period (2004-2016), all parameter combinations led to U distributions that were similar to the observed. Pearsons R reveals poor correlation between simulated and observed U values before 2003, with the best results shown by the $U_{METhigh}$ parametrization (Figure 4b). This changes for the period 2004-2016, where overall better R values were achieved, and the best results resulted from the parameterizations $U_{METlow}$, and $U_{METmean}$. The share of autotrophic nitrogen demand on total metabolic nitrogen demand $f_{AUT}$ increased for all three parametrizations (Figure 4c). The highest $f_{AUT}$ values come from the parametrization with low heterotrophic nitrogen demand ($U_{METlow}$) and vice versa.

## 4. Discussion

### 4.1 DIN Retention

We classified observed in-stream nitrogen retention $U_{obs}$ in three regimes: Regime 1 had the highest retention, a pronounced seasonal amplitude and peaked shortly before the beginning of the aquatic vegetation period, during the colder half of the year (Fig. 2a). Peaks outside the warmer half of the year were not expected as water temperature (Sherman et al., 2016) and residence time (Bertuzzo et al., 2017), considered the main drivers of in-stream DIN retention, are lower during this period. During regime 2, $U_{obs}$ gradually decreased, reduced its amplitude, and shifted its annual peak into summer, where it remained in regime 3. Higher DIN retention rates during phases of higher DIN pollution, as observed in regime 1, have already been reported by Kelly et al. (2021). They also remarked on changes in the drivers of $U_{obs}$, which reached high values even during the cold season during phases of high pollution, which agrees with our observations. We interpret regime 2 as a transition period, corresponding to the long-term improvements in wastewater treatment that followed the German reunification in 1990 and took multiple years (IKSE, 2010). During this period $NH_4$-N concentrations started to decrease during summer first, indicating changes in point source inputs (Wachholz et al., 2022). During regime 3, $U_{obs}$ shows a stable pattern with peaks coinciding with high water temperatures and residence times (Fig. 1b, c).

As neither temperature nor discharge exhibited noteworthy trends in seasonality in the Elbe during the studied period (Markovic et al., 2013; Mudersbach et al., 2016), we interpret this seasonality change as an indicator of a change in other driving processes, most likely in the abundance and metabolic activity of the responsible organisms. While the DIN input has a constant seasonal pattern, its magnitude and composition (share of $NH_4$-N) changed remarkably throughout the time series (Figure 2c). As biota, such as algae, are known to form their assemblages according to environmental factors such as light,

temperature, and nutrient availability (Snell et al., 2019), a biotic regime shift could have contributed to the $U_{obs}$ changes.

## 4.2 Metabolism

### 4.2.1 Dissolved Oxygen

Analyses of the oxygen saturation in the investigated segment of the Elbe revealed a constant saturation deficit, which was

replaced by strong seasonal pattern with pronounced super-saturation during summer (Fig. 3a) after 1990. It is understood that the oxygen budget of the Elbe after 1990 is controlled by primary production, which rapidly increased after 1990 (Lehmann and Rode, 2001; Petersen and Callies, 2002). The absence of super-saturation before 1990 could be related to high concentrations of suspended solids, which are known to limit GPP in rivers (Trentman et al., 2022) and which were observed to decrease in the Elbe following the improvements in wastewater treatment after 1990 (Hillebrand et al., 2018). However,

toxic effects from industrial waste also seem plausible, as concentrations for many organic and inorganic pollutants rapidly decreased around 1990 (Adams et al., 2001). However, a lack of primary production would not explain the significant saturation deficit of DO before 1990. From a mass balance perspective, only high rates of ER could have caused that phenomenon.

Further assessment of this requires the estimation of Elbe's metabolic processes which we carried out with a maximum

likelihood estimation.

### 4.2.2 Gross primary production and ecosystem respiration

GPP and ER estimations revealed an increase in GPP and a decrease in ER around 1990 (Fig. 3c, d). While the GPP increase was sudden and mostly affected summer peaks, ER started to decrease in the late 1980s and also shows drastically reduced

rates during the winter months. Compared to $DO_{sat}$, the timing of the GPP peak stayed similar throughout the time series (~ day 190 (July), Fig 3d), which suggests constant drivers but some limitation before 1990. It is well known that light and flow regimes control the metabolism of rivers (Bernhardt et al., 2022), so the peak during high temperature and residence times is to be expected and suggests another limiting factor before 1990 (e.g., high light attenuation from substances or sediments delivered by WWTP effluents).

The higher rates of ER before 1990 could have been supported by high loads of organic pollutants from wastewater (Adams et al., 1996). Compared to the increase in GPP, the slower ER decrease could be explained by the gradual improvements in water treatment throughout the 1990s and early 2000s (Adams et al., 2001; Wachholz et al., 2022). However, it has to be acknowledged that ER started to change before 1990. This could have been caused by an early onset of the industrial collapse of the GDR, affecting some industrial wastewater emitters, or by improved wastewater treatment before the GDR collapse.

Another interesting observation is the high winter ER before 1990. One must consider the term '*ER*' from Eq. IV contains all processes that consume DO in the balance, including nitrification. $NH_4$-N was present in high concentrations in the River Elbe before 1990 (Figure 2c). It is well known that nitrification can significantly contribute to oxygen depletion (Powers et al., 2017) and that nitrification rates during cold periods increase with $NH_4$-N concentrations (Cavaliere and Baulch, 2019). This correlates to our observation of high DIN retention rates outside the warmer half of the year, as nitrifying and denitrifying organisms also have nitrogen demands.

It has to be acknowledged that the metabolic estimations presented here are based on a few speculative assumptions. Nevertheless, the model showed over all good agreement with the observed DO data (Fig. S10). $R^2$ was generally the highest during spring (>0.8) and lowest in winter (~0.6). $R^2$ performance was similar for the period before and after 1990. Considering the root mean square error, all seasons in both periods show values between 0 and 2.5 mg $O_2$ $m^{-2}$ $d^{-1}$, with the exception of summer after 1990, where the mean root mean square error is around 7 mg $O_2$ $m^{-2}$ $d^{-1}$. This is, however, the part of the year with the highest GPP and ER values, and since the $R^2$ in this period and season is >0.7, we still consider the models performance well.

Our simple one-factor-at-a-time sensitivity analysis revealed that both GPP and ER are most susceptible to changes in k600 (Fig. S11. Since this parameter was also estimated from a hydraulic equation and could not be verified for the studied segment, this introduces an unquantified uncertainty to our results. GPP, however, was affected very little by variability in K600 compared to ER, and as similar summer GPP rates were found in the Elbe in other studies (e.g. Kamjunke et al., 2021) we assume the GPP values and long-term changes to have a high certainty. The absolute ER and k600 values for the Elbe could not be independently verified. However, the reduction of ER rates due to wastewater treatment improvements seems plausible. If we assume k600 to be solely driven by hydraulic factors, for which we found no indication of change, then an overall decrease of ER could be a likely mechanistic explanation of the observed changes in DO.

**4.3 Linking Metabolism and DIN retention**

Despite its simple nature, the metabolic N demand model showed promising results in predicting mean DIN retention rates (Figure 4a-c). Between 1979 and 1990, only the parametrization $U_{METhigh}$ was able to mimic the distribution of observed DIN

retention rates correctly. Only after 2003 did the model gain noteworthy predictive capabilities for the observed daily DIN retention rates (R ≥ 0.7) for the parametrization $U_{METlow}$ and $U_{METmean}$, with the parametrization $U_{METhigh}$ performing worse (R ≈

0.5). The overall observation of low model performance before 2003 can be interpreted as a weakened coupling between metabolic processes and in-stream DIN retention. As we hypothesized, processes such as denitrification, $NH_4$-N sorption, and nitrification can impact the DIN and oxygen balance differently, weakening the link between both elemental cycles. While we have no independent information on the rates of denitrification, $NH_4$-N sorption, and nitrification in the studied segment before 1990, we argue that the circumstances indicate that they likely were higher than after 2000. With little $NH_4$-

N present in the segment after the improvements in wastewater treatment, neither sorption nor nitrification could have occurred at high rates. Nitrification, on the other hand, is assumed to have been a major drain on the oxygen budget of the Elbe before reunification (Wachholz et al., 2024). Denitrification is assumed to be a minor part of the Elbe's DIN retention in recent years (Schulz et al., 2023), but it is known that lower DO concentrations in eutrophic rivers, as were experienced in the Elbe before 1990, can promote denitrification (Rysgaard et al., 1994). The DIN retention after 2003 is likely largely due

to assimilation by heterotrophic and autotrophic organisms and, therefore, closely linked to the metabolic processes, as indicated by the improved model performance in this period.

Our findings also have implications on how much of the metabolic nitrogen demand comes from the assimilation by autotrophic organisms. The $U_{METhigh}$ parametrization, which is the only one able to explain the high DIN retention values

before 1990 (Fig. 4a), predicts that more than 90% of DIN retention comes from heterotrophic nitrogen demands. Even if this parametrization is unrealistic and denitrification would have been a major part of the observed DIN retention, it seems likely that most DIN retention, at least before 1990, was caused by heterotrophic organisms. On the other hand, after 2003, the $U_{METlow}$ and $U_{METmean}$ show the best performance and predict that between 50% and 80% of the observed DIN retention was caused by autotrophic organisms. We interpret this as a shift between an in-stream DIN retention regime dominated by

heterotrophic activity to one dominated by autotrophic activity.

As in-stream processes are greatly influenced by the hydro-climate of the river (e.g., Bernhardt et al., 2022) and changes in its flow regime, as caused for example by dams (e.g. Aristi et al., 2014), these are competing drivers for the observed changes. At least as of 2013, no systematic trends in the Elbe's flow regime had been documented (Mudersbach et al., 2016).

The Elbe's water temperature is rising (~0.01 ° C per year) in concert with rising air temperatures, and a phase shift towards and earlier warming of two weeks has been described (Markovic et al., 2013). Based on these findings however we find it unlikely that hydro-climatic changes caused there here described changes in in-stream processes.

Our results also highlight the importance of the parameters $RQ$, $GH_{HET}$, and $C:N_{HET}$, which, within reasonable bounds, can

change heterotrophic nitrogen demand by a factor 10 (Fig. 4a). $GE_{HET}$ is often positively correlated with overall bacterial productivity or phosphorus concentrations (del Giorgio and Cole, 1998; Smith and Prairie, 2004). As both declined during

the studied period, a decline in $GE_{HET}$, as suggested by the decreasing performance of the $U_{METhigh}$ parametrization, seems plausible. For RQ, the quality of the available organic matter seems to be important, with small organic acids leading to especially high RQs (Bergren et al., 2012; Allesson et al., 2016). However, this was not assessed in this study. Overall, the

variability in the literature values for $GH_{HET}$ is ten times higher than for RQ (Tab. 1). Understanding and constraining the variability in these parameters, especially $GE_{HET}$, is therefore paramount for a better understanding of carbon, oxygen and nitrogen cycles in rivers. In our simple analysis, we ignored variability in the parameters $C:N_{AUT}$, $GE_{AUT}$, and $PQ$, which could explain the still improvable fits of the metabolic nitrogen demand model after 2003.

**4.4 Ecological implications**

Long-term changes in GPP and ER can have a multitude of ecological implications for the river segment itself or downstream ecosystems. For example, high ER and low GPP, as we observed in the Elbe before 1990, can lead to increased riverine $CO_2$ emissions (Attermeyer et al., 2021). Higher primary production in the Elbe after 1990 caused greater export of organic matter to the estuary during summer, which supports higher rates of estuarine ecosystem respiration, which in turn

decreases DO concentrations (Amann et al., 2012) and which reduces available habitats for fish species over extended stretches of the tidal zones in the estuary (Mann, 1996).

The changed seasonality of DIN retention also likely had an impact on downstream ecosystems. Even though absolute retention rates before 1990 were higher (Fig. 2a), only 20-25 % of DIN was retained during summer (Fig. 2b), when algal blooms are most likely to occur. After the German reunification, the decreased DIN load (Fig. 2c) and increased in-stream

retention led to less DIN being exported to the estuary during summer, decreasing the probability of DIN induced estuarine algal blooms (Anderson et al., 2008).

**5. Conclusions**

Our study provides valuable insights into the long-term effects of inorganic and organic pollution reduction and the tight coupling with the ecosystem functions of DIN retention and metabolism within large rivers. The shift from a heterotrophic-

dominated to an autotrophic-dominated DIN retention regime has implications for the carbon cycle and algal blooms in downstream ecosystems.

Our findings highlight the importance of considering different dimensions of integrative ecosystem functions (metabolism, DIN retention) when assessing long-term ecological changes in rivers, such as eutrophication. Dissolved oxygen concentration time series alone are sufficient to support our findings of a heterotrophic to autotrophic regime shift, but

estimates of GPP, ER, and DIN retention were required to support a quantitative assessment of the magnitude and consequences of this shift. The discovery of decoupled responses of ER and GPP to improvements in water quality offers

new insights on the time scales of aquatic ecosystem responses to changing external forcing and informs realistic estimates for the efficiency of nutrient management and the achievement of environmental objectives.

## Competing interests

The contact author has declared that none of the authors has any competing interests.

## Acknowledgments

AW was funded by the Helmholtz-International Research School "Trajectories towards Water Security" (TRACER, grand no. HIRS-0017).

## Author contribution statement

| | |
|---|---|
| Conceptualization: | all Authors |
| Methodology: | AW, JWJ |
| Data curation: | AW |
| Formal analyses and investigation: | AW, JWJ |
| Writing (original draft): | AW |
| Writing (review and editing): | AW, JWJ, DB |
| Supervision: | JWJ, DB |

## Data & Code availability

Q, DIN, temperature and DO time series can be downloaded from the FGG Elbe webportal www.elbe-datenportal.de/FisFggElbe/.

The code for the metabolism estimation can be found at github.com/alexiwach/MetabolismModelElbe. The code used for the mass balance analysis and figure creation is available from the author upon reasonable request.

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
