# Peer review of "From iron curtain to green belt: Shift from heterotrophic to autotrophic nitrogen retention in the Elbe River over 35 years of passive restoration"

_Biogeosciences, 2023_

## Referee Comment (RC1)

**Review for bg-2023-184: From iron curtain to green belt: Shift from heterotrophic to autotrophic nitrogen retention in the Elbe River over 35 years of passive restoration**

**Overall:**

The authors present estimates of DIN retention along a 100 km reach of the Elbe River across a period of dramatic reduction in industrial and wastewater processes. The authors use estimates of metabolism to partition DIN retention into autotrophic or heterotrophic compartments and craft an interesting and compelling narrative about how changes in organic pollution and nitrate result in shifting regimes of autotrophic or heterotrophic DIN uptake dominance. This work has strong potential, but I hope that the authors spend more time on developing a sturdier methodological framework before drawing their conclusions. This framework should be able to be replicated by a peer. I have listed several comments that should help them along this path. The figures are quite clear and informative—nice work!

The major comments I list will likely take considerable effort, but none are unreasonable given the work that has already been done. I caution the authors on their development of synthetic DO time series, subsequent metabolism calculations, and the ultimate uncertainty around those calculations. There are many "minor comments", some of which require more effort than others, but I regard them as not having strong influence on results or inference. Finally, I advocate for separation of Results and Discussion. I think it will help the authors synthesize their results more effectively, and will greatly help the reader in understanding the important take-homes from this work.

**Major Comments:**

2. Data and methods
This section would greatly benefit from a data collection or acquisition section that says where the data came from, the data resolution, the tools/instruments/methods used to collect/analyse it, the frequency of measurements, location of measurement, etc. The reader, even after reading the supplement, is not given this information. See for example:

L117-118: From where are these DIN data collected? Who collected them? What were the protocol? Protocols often change over such a time span, and these protocol may have differing uncertainties. It's worth mentioning and discussing.

In 2.2 Study Site, you can add also here the depth, width, other water quality parameters of interest (e.g., alkalinity, pH, phosphorus). You can further describe the changes in vegetation/trophic state over time, what is meant by the "vegetation period" that you refer to later.

2.4 Metabolism estimates and S4
A few notes on these sections and oxygen/metabolism analysis:
1) you say "light use efficiency (k600)" at the top of S4…I imagine this is just a typo. k600 is the gas exchange velocity.

2) you don't say if you used hierarchical modelling or not. How did you fit each day with the Bayesian model? A continuous time series, or in daily chunks? You should provide the equation and/or the code.

3) "On 16 % of all days, k600 was negative, and those days occurred when residence times, water temperatures were high and DOsat is > 100% (Fig. S8)"

I think this is possibly a function of the way you created your time series. The hour of peak DO strongly influences estimates of k, and you'll notice that your model is peaking after your observations. This is an important point as incorrect k strongly influences estimates of ER. Moreover, the timing of when you would expect DIN retention to be highest (high temperature, high residence time, and high GPP) are the days when your k is negative. So precisely during the period of the thing you are interested in, you have the worst estimates of metabolism. I caution the authors to be more careful here.

4) Adding to the above point, is there an inherent reason to use a sine function with a mean of 16 for phi? Perhaps a generalised additive model would be better? Probably not too important, but might help reduce your error.

5) It is well-known that there is an equifinality problem in simultaneously estimating GPP, ER, and K. One simple way to evaluate if your data have this problem is to look for collinearity in estimated ER and K. Strong linearity implies that ER and K are poorly constrained, and this should be reported. One of the ways previous authors have dealt with this issue is to hierarchically model K so that days with similar discharge have similar K. This might not be necessary for your data, but I suspect it is why you have so many negative K days.

6) An additional metric that should be considered when evaluating the fit of the Bayesian model with multiple chains is the Gelman-Rubin statistic, Rhat. That helps to diagnose your days with poor model convergence.

**2.6 Data Preparation**

L177–180: "To estimate the effects of using daily instead of hourly water temperature measurements, we calculated the mean diurnal temperature variability from 24 years of hourly water temperature in the Elbe, which is 1.1 deg C (+- 0.7). For typical $DO$, $T$, and $p$ conditions at the 180 Elbe, this can lead to a deviation in $DO_{sat}$ of a maximum of 5.4 % (see Fig. S5), which we neglect in the following analysis"

The mean over 24 years won't be informative for this analysis. I imagine, based on experience, that the temperature change in winter will be near 0, which will heavily bias your mean towards 0. Moreover, winter is when there is no DIN retention, so you're missing the effect of temperature variability during the period of most interest—summer. There can be much larger temperature changes in summer during low-flow (e.g., on the order of 5-6 degrees C), which can be up to 1 mg/L difference in $DO_{sat}$, or more like 10% change. I'm not saying you can reconcile this issue, but it may be worth considering this uncertainty in your calculations and uncertainty propagation.

**2.7 Estimating the N demand of metabolic processes**

L218–220: "As $U_{obs}$ and $U_{met}$ values have relatively high uncertainties during Regime 1 (Fig. 2; Fig. S9b), we chose a seasonality-based validation approach for the metabolic N demand model. For the $U_{obs}$ and $U_{met}$ time series, we compared the annual mean (μ), 220 the day of the peak (ϕ), and the seasonality index (SI)."

This seems a bit forced to present a good "fit". Why not just compare the model and the observations with standard RMSE, bias, etc. and then try to understand why "Regime 1" has

a poorer fit than the others? Perhaps the data is of lower quality or the simulated oxygen and subsequent metabolism data are incorrect?

**3.1 DIN retention**
There is a missed opportunity to discuss the changing peak in DIN retention from Julian day 106 to 182 (from mid-April to the beginning of July) across the study period. Considering that results and discussion are bundled in this work, why do you think this is? To me, the obvious explanation is that in Regime 3, DIN retention is controlled by GPP, which exhibits a seasonal peak around day 180–190, whereas previously retention was controlled by ER, which (typically) exhibits less seasonality. I'd advocate for separating Results and Discussion for this reason—it will allow the authors to be more synthetic in their writing, which the scope of their analysis seems to beg for.

**Figure 3**
The summary time series of ER pre-1990 highlight a likely issue with this analysis. This is a very unusual pattern for ER, even in a more heterotrophic system. What is likely happening is that you are dealing with k600 and ER equifinality during this period, and thus ER estimates are hard to trust. I again caution the authors to be careful with their metabolism analysis. It may be simpler (and instructive) to apply a hydraulic equation (Raymond et al. 2012) to estimate k600 each day, and then only estimate GPP and ER using their Bayesian approach. How much does this change your results and inference?

**Figure 4**
I really like Figure 4a, but I do not get the rationale for 4b-d. Why use the annual mean, the seasonality index, and the day of peak DIN retention as your metrics for your model? This is never clearly explained, and it seems forced. The results from Figure S9 with reported RMSE and bias seem important to include here, as well (i.e., not in the supplement). The entirety of this analysis hinges on one free parameter, the growth efficiency of heterotrophs – a relatively un-measured/unknown quantity in large rivers. What happens if ER is systematically biased in earlier years (and it likely is) in this analysis? PQ also varies greatly depending on organisms, light, nitrate, and O2 conditions, all of which are changing over time. I understand the need to simplify this analysis (and I appreciate the power of back-of-the-envelope calculation), and that perhaps this is just a first step into more detailed work, but that needs to be made more obvious. More effort needs to go into the Methods to provide a rigorous framework for the analysis set forth here. That in addition to a more fair-handed view of the uncertainty in this result.

**Minor Comments:**

L26: No Diamond et al. 2022b in references.

L28: No Ehrhardt et al., 2019 in references.

L59: GDR not defined yet.

L66: Need to defined "heterotrophic" and "autotrophic-dominated" metabolic regimes. Metabolic regime can have many interpretations; which are you referring to?

L70-71: GPP and ER not yet defined.

L74: DO not yet defined.

L82-83: "GPP in the Elbe is mostly caused by phytoplankton (Hardenbicker et al., 2014)…" Can you be more quantitative here? Moreover, If that's true, is it really "retention"? Wouldn't this just be a change in form from DIN to particulate organic nitrogen? Which is not retained, but is transported downstream to be respired and turned back into DIN?

L83: "its" is for phytoplankton or GPP here?

L85: "10%" of…total retention? What does this number refer to?

L86: "nitrification" I think a short paragraph defining the author's conception of "retention" is warranted. What role does nitrification play in retention?

L89: "DIN retention" Again, so far it sounds like autorophic DIN uptake, not retention.

L91: Is the hypothesis you are referring to the previous sentence? That sentence is not very easy to transform into a testable hypothesis as written. Please spend some more time to clarify your main question and hypothesis.

L122: "…as described in Wachholz et al. (2022)" That's fine, but you could briefly say the gap-filling method. Do the two sites have nearly identical water chemistry? Was it a linear regression?

L123: "The discharge gage is located 50 km downstream of the sampling site used to estimate DIN load (station Geesthacht)." But it's also 161km from the upstream site used to calculate load, right? That's far.

L125: "…a previous mass balance study (Ritz and Fischer, 2019)  assumed the errors to be ≤ 5% in the Elbe." Can you be more specific here? The mass balance study assumed—or calculated?—the errors for this specific section of the Elbe to be 5% from the sampling gage? Hard to know what this means without digging into the other paper.

L139: "…was computed based on Gaussian error propagation (Section S2)." The 10% error assumption for C and Q is fine if you could show (with data or reference) that it's conservative. I'm not sure it is. Wouldn't the upstream and downstream Q have different uncertainty? You mention lack of "noteworthy" tributaries, but lateral gain in flow may be a few percent as well?

L144–145: "We calculate $R_{obs}$, $RR_{obs}$, and $U_{obs}$ for both DIN and $NH_4$-N." Why? Why not also NO3?

L149: "…when a discharge mass balance was considered." I don't understand this. Don't you use the same Q for Lout and Lin? Please clarify.

L151–153: Please reformulate this sentence…I think it may have been two sentences originally.

Equation IV: "par" is not defined in the text.

L157: "…k600 is the gas exchange coefficient…" should be gas exchange "rate" or "velocity", not coefficient.

L157: "…Schmidt number…" should be Schmidt number "for oxygen".

L160: "Section S4". See above comment in major comments.

L185: "*PQ* and *RQ* describe the ratio of $O_2$ produced/ consumed per $CO_2$ consumed/ produced." Small quip here: my understanding is that PQ has units O2/CO2, whereas RQ has units CO2/O2. I don't think it matters much in this sentence, but in Equations V and VI I think it does.

L198: "…however, it is well known that the measured ER not only caused by heterotrophic bacteria." Check phrasing.

L213: "…*curve_fit…*" More detail needed here. What method does this function use? An educated reader should be able to replicate this analysis.

L251: put units of "%" on "19 to 34"

Figure 3: Increase font size, please.

L290–291: "…high internal consistency." What does this mean?

L314: "…but it is unclear how that translates to rivers." Why? It's the same process, right?

L317–319: You've mentioned this several times and I would tend to agree…are there data to support this? E.g., BOD5?

L348–349: Can you expand on this logic? Why does low $GE_{het}$ align with low $NH_4$? Wouldn't lower $NH_4$ necessitate organisms to be more efficient?

L369: "Considering our hypothesis…" What hypothesis? There doesn't seem to be one stated.

L408: "…of an autotrophic to heterotrophic regime shift…" But it's been heterotrophic the entire time, correct? Please be more clear on your use of these terms.

---

## Author Comment (AC1)

Responses to the comments

**"From iron curtain to green belt: Shift from heterotrophic to autotrophic nitrogen retention in the Elbe River over 35 years of passive restoration"**

Dear Ji-Hyung Park and reviewers,

We thank the reviewers for their extensive and instructive comments, which we relied upon to further improve the manuscript. Please find below our detailed responses to the reviewers' comments. In response to the very detailed comments, we changed our metabolism estimation method and greatly reduced the complexity of the metabolic nitrogen demand estimations to clearly reflect the uncertainty in many parameters which is well discussed in the literature. As suggested by reviewer 1 (J. Diamond), we have also separated the "Results & Discussion" section. This led to large amounts of text being recorded as "changed" in the tracked-changes version of the manuscript, even though the content was simply moved and adjusted to the standalone "Results" and "Discussion" sections. That most of these changes were editorial is further reflected by the subsection "Ecological Implications" (now 4.4) and the "Conclusions" (now 5) which remained largely unaltered. In order to help the editors and reviewers understand which sections are new, altered or just have been moved, we added comments to the tracked-changes version of the manuscript.

We look forward to receiving your decision and thank you for considering our manuscript to be published in Biogeochemistry.

Sincerely on behalf of the authors

Alexander Wachholz

**Point-by-point responses to the reviewer's comments**

**Reviewer 1:** *Jacob Diamond*

**Comment 1**
2. Data and methods
This section would greatly benefit from a data collection or acquisition section that says where the data came from, the data resolution, the tools/instruments/methods used to collect/analyse it, the frequency of measurements, location of measurement, etc. The reader, even after reading the supplement, is not given this information. See for example: L117-118: From where are these DIN data collected? Who collected them? What were the protocol? Protocols often change over such a time span, and these protocols may have differing uncertainties. It's worth mentioning and discussing.

**Answer to comment 1**

We agree with the reviewer and have added more explanation to the respective paragraph (lines 131-135) and added supplementary figure S2.

**Comment 2**

In 2.2 Study Site, you can add also here the depth, width, other water quality parameters of interest (e.g., alkalinity, pH, phosphorus). You can further describe the changes in vegetation/trophic state over time, what is meant by the "vegetation period" that you refer to later.

**Answer to comment 2**

As suggested by the reviewer, we expanded the paragraph (lines 114-121) and also changed the phrasing from vegetation period to aquatic productive period which we define in the same paragraph.

**Comment 3**
2.4 Metabolism estimates and S4
A few notes on these sections and oxygen/metabolism analysis:
1) you say "light use efficiency (k600)" at the top of S4…I imagine this is just a typo. k600 is the gas exchange velocity.

**Answer to comment 3**
Corrected as suggested by the reviewer.

**Comment 4**
2) you don't say if you used hierarchical modelling or not. How did you fit each day with the Bayesian model? A continuous time series, or in daily chunks? You should provide the equation and/or the code.

**Answer to comment 4**
As suggested by the reviewer, we added more explanation on how the model was run (lines 199 following) and made the model code available on github (github.com/alexiwach/MetabolismModelElbe).

**Comment 5**
3) "On 16 % of all days, k600 was negative, and those days occurred when residence times, water temperatures were high and DOsat is > 100% (Fig. S8)"
I think this is possibly a function of the way you created your time series. The hour of peak DO strongly influences estimates of k, and you'll notice that your model is peaking after your observations. This is an important point as incorrect k strongly influences estimates of ER. Moreover, the timing of when you would expect DIN retention to be highest (high temperature, high residence time, and high GPP) are the days when your k is negative. So precisely during the period of the thing you are interested in, you have the worst estimates of metabolism. I caution the authors to be more careful here.

**Answer to comment 5**
We thank the reviewer for his detailed considerations. As suggested, we have revised our metabolism analyses (lines 199 following) and added a more thorough discussion of the results (lines 488-503).

**Comment 6**
4) Adding to the above point, is there an inherent reason to use a sine function with a mean of 16 for phi? Perhaps a generalised additive model would be beler? Probably not too important, but might help reduce your error.

**Answer to comment 6**

We agree with the reviewer that there is no inherent reason to use a sine wave function. However, we believe that our approach produces overall good results and have revised figure S7 to show this in a more systematic way.

**Comment 7**
5) It is well-known that there is an equifinality problem in simultaneously estimating GPP, ER, and K. One simple way to evaluate if your data have this problem is to look for collinearity in estimated ER and K. Strong linearity implies that ER and K are poorly constrained, and this should be reported. One of the ways previous authors have dealt with this issue is to hierarchically model K so that days with similar discharge have similar K. This might not be necessary for your data, but I suspect it is why you have so many negative K days.

**Answer to comment 7**
We again thank the reviewer for his in-depth comments and refer to our answer to comment 5.

**Comment 8**
6) An additional metric that should be considered when evaluating the fit of the Bayesian model with multiple chains is the Gelman-Rubin statistic, Rhat. That helps to diagnose your days with poor model convergence.

**Answer to comment 8**
We fully agree with the reviewer. We have changed our model framework, as described in our answer to comment 5.

**Comment 9**
2.6 Data Preparation
L177–180: "To estimate the effects of using daily instead of hourly water temperature measurements, we calculated the mean diurnal temperature variability from 24 years of hourly water temperature in the Elbe, which is 1.1 deg C (+- 0.7). For typical DO, T, and p conditions at the 180 Elbe, this can lead to a deviation in DOsat of a maximum of 5.4 % (see Fig. S5), which we neglect in the following analysis"
The mean over 24 years won't be informative for this analysis. I imagine, based on experience, that the temperature change in winter will be near 0, which will heavily bias your mean towards 0. Moreover, winter is when there is no DIN retention, so you're missing the effect of temperature variability during the period of most interest—summer. There can be much larger temperature changes in summer during low-flow (e.g., on the order of 5-6 degrees C), which can be up to 1 mg/L difference in DOsat, or more like 10% change. I'm not saying you can reconcile this issue, but it may be worth considering this uncertainty in your calculations and uncertainty propagation.

**Answer to comment 9**
We agree with the reviewer and have revised our consideration of the uncertainties. This is now briefly described in the lines 204 and following, and more in depth in the Section S4 of the supporting information.

**Comment 10**
2.7 Estimating the N demand of metabolic processes
L218–220: "As Uobs and Umet values have relatively high uncertainties during Regime 1 (Fig. 2; Fig. S9b), we chose a seasonality-based validation approach for the metabolic N demand model. For the Uobs and Umet time series, we compared the annual mean ($\mu$), 220 the day of the peak ($\phi$), and the seasonality index (SI)."

This seems a bit forced to present a good "fit". Why not just compare the model and the observations with standard RMSE, bias, etc. and then try to understand why "Regime 1" has a poorer fit than the others? Perhaps the data is of lower quality or the simulated oxygen and subsequent metabolism data are incorrect?

**Answer to comment 10**
We agree with the reviewer and have revised our metabolic nitrogen demand modeling approach (lines 252 and following).

**Comment 11**
3.1 DIN retention
There is a missed opportunity to discuss the changing peak in DIN retention from Julian day 106 to 182 (from mid-April to the beginning of July) across the study period. Considering that results and discussion are bundled in this work, why do you think this is? To me, the obvious explanation is that in Regime 3, DIN retention is controlled by GPP, which exhibits a seasonal peak around day 180–190, whereas previously retention was controlled by ER, which (typically) exhibits less seasonality. I'd advocate for separating Results and Discussion for this reason—it will allow the authors to be more synthetic in their writing, which the scope of their analysis seems to beg for.

**Answer to comment 11**
As suggested by the reviewer, we separated the results and discussion sections in the paper. We agree with his interpretation and have made this clearer in the manuscript (lines 438 and following).

**Comment 12**
Figure 3
The summary time series of ER pre-1990 highlight a likely issue with this analysis. This is a very unusual palern for ER, even in a more heterotrophic system. What is likely happening is that you are dealing with k600 and ER equifinality during this period, and thus ER estimates are hard to trust. I again caution the authors to be careful with their metabolism analysis. It may be simpler (and instructive) to apply a hydraulic equation (Raymond et al. 2012) to estimate k600 each day, and then only estimate GPP and ER using their Bayesian approach. How much does this change your results and inference?

**Answer to comment 12**
As suggested by the reviewer, we re-ran the analysis with hydraulically estimated k600 values (see our answer to comment 5). This has in fact removed the unusual ER seasonality from the results.

**Comment 13**
Figure 4
I really like Figure 4a, but I do not get the rationale for 4b-d. Why use the annual mean, the seasonality index, and the day of peak DIN retention as your metrics for your model? This is never clearly explained, and it seems forced. The results from Figure S9 with reported RMSE and bias seem important to include here, as well (i.e., not in the supplement). The entirety of this analysis hinges on one free parameter, the growth efficiency of heterotrophs – a relatively untimeasured/unknown quantity in large rivers. What happens if ER is systematically biased in earlier years (and it likely is) in this analysis? PQ also varies greatly depending on organisms, light, nitrate, and O2 conditions, all of which are changing over time. I understand the need to simplify this analysis (and I appreciate the power of back-ofthe-envelope calculation), and that perhaps this is just a first step into more detailed work, but that needs to be made more obvious. More effort needs to go into the Methods to

provide a rigorous framework for the analysis set forth here. That in addition to a more fairhanded view of the uncertainty in this result.

**Answer to comment 13**

We thank the reviewer for this very detailed insight. Inspired by this comment, we changed our metabolic nitrogen demand framework (see our answer to comment 10). The analysis is now simpler and reflects the uncertainties of the parameters clearly.

**Minor comments**

L26: No Diamond et al. 2022b in references.

 -> corrected

L28: No Ehrhardt et al., 2019 in references.

-> corrected

L59: GDR not defined yet.

-> corrected

L66: Need to defined "heterotrophic" and "autotrophic-dominated" metabolic regimes. Metabolic regime can have many interpretations; which are you referring to?

-> We agree with the reviewer and have added the definition in the lines 69, 70.

L70-71: GPP and ER not yet defined.

-> corrected

L74: DO not yet defined.

-> corrected

L82-83: "GPP in the Elbe is mostly caused by phytoplankton (Hardenbicker et al., 2014)…"
Can you be more quantitative here? Moreover, If that's true, is it really "retention"?
Wouldn't this just be a change in form from DIN to particulate organic nitrogen? Which is not retained, but is transported downstream to be respired and turned back into DIN?

-> We agree and have added an explicit definition at lines 39, 40.

L83: "its" is for phytoplankton or GPP here?

-> Clarified, now line 87.

L85: "10%" of…total retention? What does this number refer to?

-> We have switched to a more recent reference and used to a qualitative statement (line 89).

L86: "nitrification" I think a short paragraph defining the author's conception of "retention" is warranted. What role does nitrification play in retention? Added explicit definition at L38,39

-> We agree and have added an explicit definition of retention including all considered processes at lines 39, 40.

L89: "DIN retention" Again, so far it sounds like autorophic DIN uptake, not retention.

-> See the previous answer.

L91: Is the hypothesis you are referring to the previous sentence? That sentence is not very easy to transform into a testable hypothesis as written. Please spend some more time to clarify your main question and hypothesis.

-> We reformulated the paragraph (lines 94-96) and explicitly formulated a hypothesis.

L122: "…as described in Wachholz et al. (2022)" That's fine, but you could briefly say the gap-filling method. Do the two sites have nearly identical water chemistry? Was it a linear regression?

-> We thank the reviewer for the comment and described the filling in more detail in the lines 141, 142.

L123: "The discharge gage is located 50 km downstream of the sampling site used to estimate DIN load (station Geesthacht)." But it's also 161km from the upstream site used to calculate load, right? That's far.

-> We thank the reviewer for spotting this ambiguity. We used two gages and revised the text in the "two station mass-balance" section (lines 138 and following) accordingly.

L125: "…a previous mass balance study (Ritz and Fischer, 2019) assumed the errors to be ≤ 5% in the Elbe." Can you be more specific here? The mass balance study assumed—or calculated?—the errors for this specific section of the Elbe to be 5% from the sampling gage? Hard to know what this means without digging into the other paper.

-> This is now described in lines 145, 146,
L139: "…was computed based on Gaussian error propagation (Section S2)." The 10% error assumption for C and Q is fine if you could show (with data or reference) that it's conservative. I'm not sure it is. Wouldn't the upstream and downstream Q have different uncertainty? You mention lack of "noteworthy" tributaries, but lateral gain in flow may be a few percent as well?

-> We agree and have added some references for the C and Q (lines 144 and following) errors and added an estimation on the lateral groundwater inflow (lines 115 and following).

L144–145: "We calculate Robs, RR obs, and Uobs for both DIN and NH4-N." Why? Why not also NO3?

-> As we did not analyze the retention of ammonium, we removed the phrase.

L149: "…when a discharge mass balance was considered." I don't understand this. Don't you use the same Q for Lout and Lin? Please clarify.

-> We thank the reviewer for spotting this ambiguity. We used two gages and revised the text in the "two station mass-balance" section (lines 138 and following) accordingly.

L151–153: Please reformulate this sentence…I think it may have been two sentences originally.

-> Corrected.

Equation IV: "par" is not defined in the text.

-> We changed the terminology to photosynthetic photon flux density (PPFD) which is defined in lines 186, 187.

L157: "…k600 is the gas exchange coefficient…" should be gas exchange "rate" or "velocity", not coefficient.

-> Corrected.

L157: "…Schmidt number…" should be Schmidt number "for oxygen".

-> Corrected.

L160: "Section S4". See above comment in major comments.

L185: "PQ and RQ describe the ratio of O2 produced/ consumed per CO2 consumed/ produced." Small quip here: my understanding is that PQ has units O2/CO2, whereas RQ has units CO2/O2. I don't think it malers much in this sentence, but in Equations V and VI I think it does.

-> We thank the reviewer for this error and have corrected the equations, including equation VIII accordingly.

L198: "…however, it is well known that the measured ER not only caused by heterotrophic bacteria." Check phrasing.
-> Corrected, now line 242.

L213: "…curve_fit…" More detail needed here. What method does this function use? An educated reader should be able to replicate this analysis.

-> This sentence was removed from the manuscript. When referring to software implementations at other parts of the manuscript, we have added more detail and in case of the metabolism estimation made the code available on github.

L251: put units of "%" on "19 to 34"

-> Done.

Figure 3: Increase font size, please.

-> Done.

L290–291: "…high internal consistency." What does this mean?

-> We have removed that statement.

L314: "…but it is unclear how that translates to rivers." Why? It's the same process, right?

-> We agree and have removed that statement.

L317–319: You've mentioned this several times and I would tend to agree…are there data to support this? E.g., BOD5?

-> We agree with the reviewer but this data is not available for the time series.

L348–349: Can you expand on this logic? Why does low GEhet align with low NH4? Wouldn't lower NH4 necessitate organisms to be more efficient?

-> We have removed this statement from the manuscript.

L369: "Considering our hypothesis…" What hypothesis? There doesn't seem to be one stated.

-> We now explicitly state a hypothesis in lines 94-96.

L408: "…of an autotrophic to heterotrophic regime shift…" But it's been heterotrophic the entire time, correct? Please be more clear on your use of these terms.

-> We have clarified the terminology throughout the manuscript, e.g. at lines 69-70, and 356 following.

**Reviewer 2:** *Anonymus*

**Comment 1**
Methodological details: Given the importance of long-term monitoring data (particularly DIN and DO), the Methods descriptions lack detailed information about data sources and processing. First, it is not clear how data sets for different stations and periods were collected and controlled for QA/QC. Second, it would help other researchers understand the principle of the mass-balance approach, as described in Intro (L 93-94), and replicate modeling procedures, if the principle, together with model components, is explained in more detail in sections 2.3-2.4 & 2.7. In particular, in section 2.7, it is hard to follow up the logic for translating PQ and RQ into Uaut and Uhet. Any empirical data available to validate the relationships?

**Answer to comment 1**
We thank the reviewer for this detailed comment and have made multiple changes to the manuscript. First, we have expanded the paragraphs describing data, sampling, and the layout of the two-station mass balance (lines 130 and following). We have also greatly reconsidered the metabolic nitrogen demand model (lines 252-263) and have used more literature data for the model parametrizations.

**Comment 2**

How N retention mechanisms shift in response to altering pollution regimes?: First, lacking definitions. Lines (L) 41-49 would be a good place where heterotrophic vs. autotrophic N retention can be defined. Please expand your discussion (section 3.3) to articulate specific heterotrophic and autotrophic processes linked to N retention based on these definitions. For instance, which in-stream DIN retention processes are responsible for the retention by autotrophs (UAUT) and heterotrophs (UHET), as described in L 351-. In addition, please elaborate why you "speculate that the ER during regime 3 depended more on autochthonous organic matter production from phytoplankton (L 375-376). Specifically, I wondered whether (and how) autotrophic N uptake had been subdued during the earlier high-pollution phase. The "power" shift needs to be explained in terms of specific metabolic processes.

**Answer to comment 2**

We agree with the reviewer and have revised the manuscript at multiple locations. First, we included a definition of the responsible processes in lines 39-40. We explained the processed included and excluded from the metabolic nitrogen demand lines 263-265. The statement in lines 531- was removed. We explain the limitation of GPP before 1990 in lines 458 and following. In lines 523 and following we name the specific metabolic processes.

**Comment 3**

Discussion on the relative importance of other factors: As the authors focus on presenting key modeling results in R & D, more discussion is required to provide a more balanced view on the relative importance of key factors involved in the metabolic shift over the different stages of pollution. For instance, don't we need to consider decadal climatic variations (like droughts or wetter climates prevailing in certain phases) or newly constructed (or decommissioned) dams and weirs as the competing or overriding factors driving the presented metabolic shift?

**Answer to comment 3**

We thank the reviewer for this important comment and have expanded the discussion in lines 532-537. In lines 111-112 we added the information about dams.

**Comment 4**

L 29-31: It would be also helpful for readers if some global estimates (often much larger than 30%) of N retention in inland waters (I.W.) are provided. For instance, a biogeochemistry textbook (Schlesinger & Bernhardt, 3rd) offers a global budget: 47.8 Tg N, out of 118 Tg N entering I.W., reaches the oceans.

**Answer to comment 4**

As suggested by the reviewer, we have adjusted the statement and replaced the reference (lines 30, 31).

**Comment 5**

L 85- (hypothesis): lower NN4 concentrations were linked to reduced nitrification. However, lowered concentrations may simply reflect a reduction in N release from the sources.

**Answer to comment 5**

We agree with the reviewer that the framing was confusing and have adjusted the paragraph (now line 91).

**Comment 6**

L 121: Given the long period, this gap filling needs to be validated by comparing data from the two stations for the two periods (gap-filled vs. pre- and/or post-gap periods).

**Answer to comment 6**

We agree with the reviewer and have expanded our explanation thereof (lines 141-143) and show the raw data in Figure S2 in the supporting information.

**Comment 7**

L 154-160: Many typos (er, gpp, K600). Please pay attention to the correct or usual acronyms (lower- vs. upper-case letters)

**Answer to comment 7**

We thank the reviewer for spotting these errors and have corrected them.

**Comment 8**

Fig.4: Hard to compare color-coded lines and symbols in three plots. Please use different colors in (b) not to the two data as Uhet AND Uaut.

**Answer to comment 8**

We agree with the reviewer and have revised figure 4 completely.

---

## Referee Report (RR1)

**Review for revisions to bg-2023-184: From iron curtain to green belt: Shift from heterotrophic to autotrophic nitrogen retention in the Elbe River over 35 years of passive restoration**

**Overall:**

I thank the authors for addressing all the reviewer comments in a clear manner. While there are some grammatical issues, small typos, and awkward wordings throughout that could use another review before final submission, I find the technical work to be sound.

**Major Comments:**

Figure 4 and associated approach: This I think is the weakest point in the manuscript, but it's not a deal-breaker. The figure and approach work as is but could be made more robust. You mention in the Discussion that PQ, $GE_{auto}$, and $C:N_{auto}$ also could be (and are) changing over time, but instead only focus your uncertainty on the respiratory side of things. Please make clearer why you made this choice. Instead of picking three extreme parameter values, couldn't you use a similar approach of ML to fit the parameters? What causes $GE_{het}$ and RQ to change? What direction would you expect them to evolve in given your understanding of the system? Reporting this would strengthen the results of the manuscript and be useful for our overall understanding riverine C and N functioning.

Using Pearson's "r" without showing the actual scatter plots seems to obscure the results – we never actually see the time series or data from the $U_{met}$ plots. That was a visually compelling result in the previous version and I'd opt to bring it back in somehow. For example, you could choose the best parameter set for each period and then show the time series of $U_{aut}$ and $U_{het}$ in panel C (as from the previous version) and add the mean $f_{aut}$ in text above each delineated period (like in Figure 2). I think you should leverage the effort you put into this and the large amount of data to make this result more compelling.

**Minor Comments:**

L89–91: First, I recommend removing the language "a natural condition" as it is meaningless here. Why do you expect weakened coupling of metabolism and DIN retention during the high pollution phase? I recommend rethinking and rephrasing this hypothesis and including at least one testable prediction from your hypothesis after this sentence. For example, "Based on this hypothesis, we predicted that respiratory processes would explain the majority of DIN retention in the high pollution period."

Figure 3: In the figure caption you mention the GPP/ER ratio, but I do not think it is shown in the figure, or mentioned in the text.

In line with the above, I recommend reporting NEP and its change over time. This value is important for understanding the trophic state and "metabolic regime" of the system.

---

## Author Response (AR2)

Dear Ji-Hyung Park and Jacob Diamond,

We thank you for the overall positive assessment of our revised manuscript and the again very detailed and instructive comments. Please find below our detailed responses to the reviewers' comments.

on behalf of the authors,
Alexander Wachholz

**Response to the reviewers comments:**

**Reviewer 1:** *Jacob Diamond*

**Comment 1:** Figure 4 and associated approach: This I think is the weakest point in the manuscript, but it's not a deal-breaker. The figure and approach work as is but could be made more robust. You mention in the Discussion that PQ, GEauto, and C:Nauto also could be (and are) changing over me, but instead only focus your uncertainty on the respiratory side of things. Please make clearer why you made this choice. Instead of picking three extreme parameter values, couldn't you use a similar approach of ML to fit the parameters? What causes GEhet and RQ to change? What direction would you expect them to evolve in given your understanding of the system? Reporting this would strengthen the results of the manuscript and be useful for our overall understanding riverine C and N functioning.

**Response to comment 1:** We agree with the reviewer and have made the following changes to the manuscript:
- In lines 248, 294, we stated why we used this specific PQ, Geauto and C:Nauto values
- In lines 442-446, we discuss the drivers of GEhet and RQ in the context of our analysis

**Comment 2:** Using Pearson's "r" without showing the actual scatter plots seems to obscure the results – we never actually see the me series or data from the Umet plots. That was a visually compelling result in the previous version and I'd opt to bring it back in somehow. For example, you could choose the best parameter set for each period and then show the me series of Uaut and Uhet in panel C (as from the previous version) and add the mean faut in text above each delineated period (like in Figure 2). I think you should leverage the effort you put into this and the large amount of data to make this result more compelling.

**Response to comment 2:** We thank the reviewer for this constructive comment and have revised Figure 4 as as suggested.

**Comment 3:** L89–91: First, I recommend removing the language "a natural condition" as it is meaningless here. Why do you expect weakened coupling of metabolism and DIN retention during the high pollution phase? I recommend rethinking and rephrasing this hypothesis and including at least one testable prediction from your hypothesis a er this sentence. For example, "Based on this

hypothesis, we predicted that respiratory processes would explain the majority of DIN retention in the high pollution period."

**Response to comment 3:** We agree with the reviewer and have rephrased the hypothesis (lines 91-93).

**Comment 4:** Figure 3: In the figure caption you mention the GPP/ER ra o, but I do not think it is shown in the figure, or mentioned in the text. In line with the above, I recommend reporting NEP and its change over me. This value is important for understanding the trophic state and "metabolic regime" of the system.

**Response to comment 4:** We agree with the reviewer and have adjusted the caption of figure 3. The NEP is reported in the lines 305-308.